# Optimization of Spray-Drying Process of *Jerusalem artichoke* Extract for Inulin Production

**DOI:** 10.3390/molecules24091674

**Published:** 2019-04-29

**Authors:** Zhenzhou Zhu, Mailing Wu, Jie Cai, Shuyi Li, Krystian Marszałek, Jose M. Lorenzo, Francisco J. Barba

**Affiliations:** 1College of Food Science and Engineering, Wuhan Polytechnic University, Wuhan 430023, China; zhenzhouzhu@126.com (Z.Z.); wml4568@163.com (M.W.); caijievip@whpu.edu.cn (J.C.); 2Prof. Wacław Dąbrowski Institute of Agricultural and Food Biotechnology, Department of Fruit and Vegetable Product Technology, 36 Rakowiecka St., 02-532 Warsaw, Poland; krystian.marszalek@ibprs.pl; 3Centro Tecnológico de la Carne de Galicia, Avda. Galicia No. 4, Parque Tecnológico de Galicia, San Cibrao das Viñas, 32900 Ourense, Spain; 4Nutrition and Food Science Area, Preventive Medicine and Public Health, Food Science, Toxicology and Forensic Medicine Department, Universitat de València, Faculty of Pharmacy, Avda. Vicent Andrés Estellés, s/n, 46100 Burjassot, València, Spain

**Keywords:** inulin, response surface methodology, spray-drying, *Jerusalem artichoke*

## Abstract

Jerusalem artichoke is an important natural matrix for inulin production. In this experiment, response surface methodology (RSM) was employed to optimize the spray-drying parameters in order to determine the maximal inulin yield. For this study, three independent variables (heating temperature (Tª, 110–120 °C), creep speed (V, 18–22 rpm) and pressure (P, 0.02–0.04 MPa)) were used in the experimental design. Using the Box–Behnken design, the optimal parameters obtained were: drying temperature 114.6 °C, creep speed 20.02 rpm, and pressure: 0.03 MPa. The inulin yield, water content and particle size of inulin obtained by spray-drying and freeze-drying were compared. In this regard, the spray-dried inulin consisted of a white powder having a fine particle size, and the freeze-dried inulin had a pale-yellow fluffy floc. On the other hand, the drying methods had a great influence on the appearance and internal structure of inulin powder, since the spray-dried inulin had a complete and uniform shape and size, whereas the freeze-dried inulin had a flocculated sheet structure. The analysis showed that the spray-drying led to a higher inulin yield, lower water content and better surface structure than freeze-drying.

## 1. Introduction

*Jerusalem artichoke* is widely planted in northwest China due to its ability to be cold resistant, containing in its tubers ≈14–19% inulin [1,2]. Inulin is a natural stored carbohydrate, which consists of 2 to 70 fructose repeating units connected by β-(2,1)-d-fructosyl-fructose bonds [3,4,5,6]. Inulin is now widely used in food due to its unique functional properties [7,8]. For example, inulin is now used as a sugar substitute since it does not cause blood sugar fluctuations [9]. The fructose syrup formed after the degradation of inulin can promote the growth of beneficial bacteria, especially bifid bacteria with health and anti-cancer effects in the large intestine [10,11]. Fructose syrup also has a positive effect on blood sugar and fat reduction, as well as on the bioavailability and immunomodulation of minerals. 

The degree of polymerization (DP) and molecular weight of inulin molecules differs according to the plant material used, with inulin of DP <10 being known as fructo-oligosaccharides or short-chain inulin, and inulin of DP > 23 called polyfructose or long-chain inulin [3,12]. Moreover, as a non-digestible carbohydrate, inulin can be also used as a fat substitute for food [13].

Inulin is mostly sold in powder form for easier handling, transportation, storage and consumption [14]. The drying technology applied to obtain inulin powder is very important in determining its quality. Currently, the drying methods for inulin preparation are mainly freeze-drying and spray-drying [15]. Freeze-drying includes sublimation and desorption, and usually takes two or three days, obtaining a porous product and loose structure [16]. Freeze-drying is relatively demanding for packaging and storage, since the product absorbs moisture and can be easily oxidized when exposed to air [17,18]. Moreover, it is a process with high production cost and limited efficiency. Therefore, the most common drying method used in food industry is spray-drying mainly due to its high cost-effectiveness and high flexibility [19]. 

Spray-drying uses a nebulizer to disperse the liquid into fine droplets and rapidly evaporates the solvent in a hot drying medium to form a dry powder product. Custom production can be achieved by varying parameters such as temperature, viscosity, feed rate or atomization pressure [20]. The particle size of the resulting product has a uniform size and good sphericity. For instance, in a previous study, the properties of Jerusalem artichoke pectin using different drying methods (freeze-drying, spray-drying and vacuum drying) were evaluated, and the authors obtained the best conditions after spray-drying, leading to pectin with the highest strength [21].

Therefore, the purpose of this study was to optimize the spray-drying process parameters using response surface methodology in order to obtain the maximal inulin yield. Moreover, the effect of different drying technologies on morphological properties of inulin were evaluated and compared.

## 2. Results and Discussion

The extraction yield of this study under the applied extraction conditions (ultrasound power of 120 W, temperature of 70 °C) was determined as ~9.5%.

### 2.1. Optimization of Spray-Drying Process by Response Surface Methodology

The test design of spray-drying and the results obtained after using response surface methodology to optimize inulin extraction are shown in Table 1. Moreover, the ANOVA analysis is listed in Table 2.

The test results in Table 1 are regression fitted, and the final equation for obtaining the coding factor by data analysis is shown in Equation (1). The regression of the actual factor is shown in Equation (2).

(1)$$Y=7.98-0.04\times A-0.081\times B-0.57\times C-0.31\times AB+0.16\times AC-0.26\times BC-0.66\times A^{2}-0.84\times B^{2}-0.99\times C^{2}$$

(2)$$y=-496.291+6.554\times X_{1}+12.281\times X_{2}+422.825\times X_{3}-0.031\times X_{1}X_{2}+3.250\times X_{2}X_{3}-13\times X_{1}X_{3}-0.026\times X_{1}{}^{2}-0.210\times X_{2}{}^{2}-9895\times X_{3}{}^{2}$$

The overall model *p*-value is 0.0008 < 0.01, demonstrating that the regression equation is very significant, and the "Lack of fitting *p*-value" is 0.4645 > 0.05, indicating that the lack of fitting is not significant. It shows that the fitting degree of the equation is relatively good. Regression analysis of the experimental regression equation (Table 2) showed that A^2^, B^2^ and C^2^ terms had a very significant effect on inulin yield. The pressure had a significant effect on inulin yield (*p* < 0.05). Moreover, the influence of the factors can be ordered according to the importance in the following row: heating temperature < creep speed < pressure.

Figure 1 provides a good view of the contour plot and the 3D response surface profiles, clearly showing the effect of the interaction of the three factors on the response. Figure 1A–C shows the effects of temperature, creep speed, and pressure on inulin yield. When the temperature increased from 110 °C to 120 °C, the yield was observed to first increase, and afterwards decrease. This fact can be attributed to an enhanced drying process when heating temperature was increased. However, it can also lead to the decomposition of inulin molecules. In addition, the increase of creep speed and pressure led to the same trend regarding inulin yield variation.

According to the experimental data and model analysis, the maximized inulin yield, could be achieved under the optimal conditions: heating temperature of 114.6 °C, creeping speed of 20.02 rpm, and pressure of 0.03 MPa, leading to an inulin yield of 8.52%. To verify the availability and reliability of the regression model obtained in the design of the response surface test, the above-mentioned optimal preparation parameters were used for the verification test. For the convenience of operation, the heating temperature was set to 115 °C, the creeping speed was 20 rpm, the pressure was 0.03 MPa, and three assays were performed in parallel. In these verification experiments, the inulin yield was 8.65 ± 0.69%, which is close to the predicted value, and indicates that the equation fits well with the real situation.

### 2.2. Comparison between Spray-Drying and Freeze-Drying

The results of test indexes of inulin obtained from spray-drying and freeze-drying are presented in Table 3.

As it is clearly shown in Table 3, the inulin yield obtained after applying spray-drying is higher than that obtained after freeze-drying. This may be due to the large amount of inulin being adhered to the ware after lyophilization, resulting in loss of inulin. However, the difference between the moisture contents is not significant. As it is shown in Figure 2a,b, the spray-dried inulin consisted of a white powder having a fine particle size, and the freeze-dried inulin had a pale-yellow fluffy floc. A similar phenomenon was also observed after drying loach peptides, where spray-drying led to a less colored peptide powder compared to freeze-drying [22].

The SEM images of inulin from different drying methods are shown in Figure 3. As can be seen in the figure, drying methods had a great influence on the appearance and internal structure of inulin powder. The spray-dried inulin had a complete and uniform shape and size, no obvious adhesion, and a granular morphology, but the surface had concave folds, which is mainly caused by the different drying strength of the different parts of the droplet during the spray-drying process. During the spray-drying process, the material is dispersed into tiny droplets by the atomizer, quickly contacts with the hot air, and is dried into a powder in a short time. The rapidly increasing surface tension and the reduced water diffusion rate led to the formation of dent wrinkle and spherical surface. The freeze-dried inulin had a more spherical structure compared to the spray-dried inulin, probably due to the relatively mild change during the freezing process.

## 3. Materials and Methods 

### 3.1. Materials and Chemicals

Fresh Jerusalem artichoke was purchased from a local supermarket of Wuhan, Hubei, China. The other chemicals used in the present study were of analytical grade and were purchased from Sinopharm Chemical Reagent Co., Ltd (Shanghai, China). 

### 3.2. Preparation of Jerusalem artichoke Extract

One kilogram of fresh Jerusalem artichoke root (washed using tap water and sliced prior its use) and 5 liters of tap water were added into an ultrasonic circulating extraction equipment (TGCXZ-10B, frequency 59 kHz, up to1000 W power, Beijing Hong Xiang Long Co., Ltd, Beijing, China) equipped with an ultrasound horn-type probe of 20 mm diameter. The ultrasound power was set at 120 W and the extraction was carried out at a temperature of 70 °C for 40 min [23]. The extract was initially filtered to remove the pulp, then passed through a 1 μm microfiltration membrane to remove large particles of tissue, and finally filtered with a 50 kDa membrane to obtain the permeate [2], which was stored at −20 °C until further use.

### 3.3. Preparation of Inulin Powder

Inulin powder was prepared using both spray-drying and freeze-drying. Response surface methodology (RSM) was used to optimize the spray-drying parameters. The coding and actual level of independent variables of the process are shown in Table 4. Based on previous single factor experiments, the three independent variables used in the experimental design were heating temperature (Tª, 110–120 °C), creep speed (V, 18–22 rpm) and pressure (P, 0.02–0.04 MPa). According to Box–Behnken design, the experimental runs performed are listed in Table 1.

The test data was statistically analyzed using Design-Expert 7.0.0 (Stat Ease Inc., Minneapolis, MN, USA). The inulin yield (Y) was fitted to a quadratic regression model for response surface analysis. As shown in Equation (3):(3)$$Y=\beta_{0}+\beta_{1}A+\beta_{2}B+\beta_{3}C+\beta_{4}AB+\beta_{5}AC+\beta_{6}BC+\beta_{7}A^{2}+\beta_{8}B^{2}+\beta_{9}C^{2}$$where A, B and C correspond to the coded independent variables, namely heating temperature, creep speed and pressure. The β_0_ value represents the corresponding regression coefficient. The experiment was randomized to maximize the effect of unexplained variability on observed responses due to exogenous factors.

Five hundred milliliters of clarified inulin juice were used for the drying experiment with a spray-dryer (YC-2000), purchased from Ningbo Haoyu Instrument Co., Ltd (Ningbo, Zhejiang, China). The sample solution was pumped through a nozzle to the countercurrent drying chamber using a peristaltic pump. Each solution of inulin was preheated to a certain temperature before each feeding (inlet temperature is higher than 60 °C, and outlet temperature > 50 °C). Spray-drying parameters for maximal inulin yield were obtained using a response surface methodology assay.

As a control sample, 500 mL of inulin juice were used for freeze-drying. The juice was pre-frozen in a refrigerator at −20 °C for over 24 h, and then freeze-dried at −50 °C for 72 h, using an ALPHA 2-4 LD plus freeze dryer (Martin Christ Gefriertrocknungsanlagen GmbH, Oster ode am Harz, Germany) [14]. 

### 3.4. Characterization and Analysis

#### 3.4.1. Determination of Inulin Yield

The yield of inulin was calculated as follows: (4)$$Y=\frac{m_{p}}{M}\times 100\%$$where m_p_ (kg) is the mass of inulin powder obtained from 5 L extract, M (kg) is the mass of Jerusalem artichoke used in the extraction to obtain 5 L of extract. 

#### 3.4.2. Determination of Water Content

The water content in the inulin solution was calculated using the mass difference method. Samples were dried in an oven at 105 °C for a certain period of time and then weighed within 30 seconds. The above procedure was repeated until weight was constant (the difference in weight of two consecutive weightings was less than 2 mg). The water content (W) of the sample was calculated according to Equation (5):(5)$$W=\frac{m-m_{0}}{m}\times 100\%$$where m (kg) is the mass fresh material, m_0_ (kg) is the mass of dried material. 

#### 3.4.3. Analysis of Particle Size and Surface Structure of Inulin Powder

The particle size of inulin powder was measured using a Malvern Zen 3600 Zeta sizer instrument (Malvern Instrument, Malvern, UK). The surface structure of inulin obtained by both freeze-drying and spray-drying was analyzed using scanning electron microscopy (SEM), S-300N (Japan Hitachi, Japan) [12,14,15,24].

### 3.5. Statistical Analysis

An ANOVA analysis was performed to evaluate the differences between freeze-drying and spray-drying. Differences at *p* < 0.05 were considered to be significant.

## 4. Conclusions

Inulin powder was prepared through spray-drying and freeze-drying. The results showed that inulin yield firstly increased when heating temperature, creeping speed and pressure were increased, but then decreased. After applying response surface methodology, the maximal inulin yield was 8.52%, which was obtained under the optimal conditions of heating temperature ≈114.6 °C, creeping speed of 20.02 rpm, and pressure of 0.03 MPa. The inulin yield obtained from spray-drying was slightly higher since loss of inulin may occur after lyophilization during the freeze-drying process. The moisture content of inulin—which is related to the shelf-life—obtained from both processes did not show any significant differences. Freeze-drying resulted in a more spherical structure while spray-drying led to fine inulin particles.

## Figures and Tables

**Figure 1 molecules-24-01674-f001:**
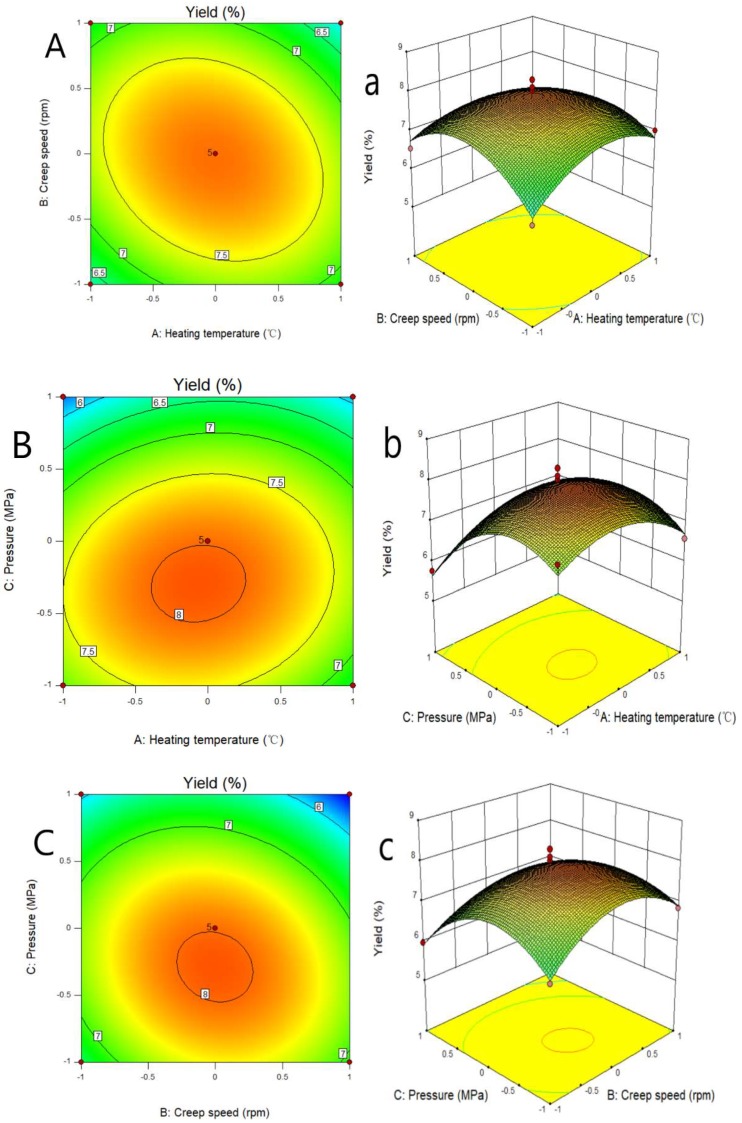
Effect of interaction of various factors on inulin yield.

**Figure 2 molecules-24-01674-f002:**
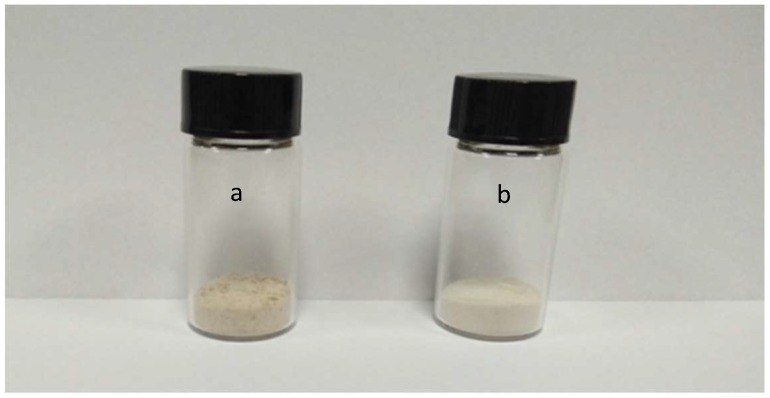
(**a**) Freeze-dried inulin sample and (**b**) spray-dried inulin sample.

**Figure 3 molecules-24-01674-f003:**
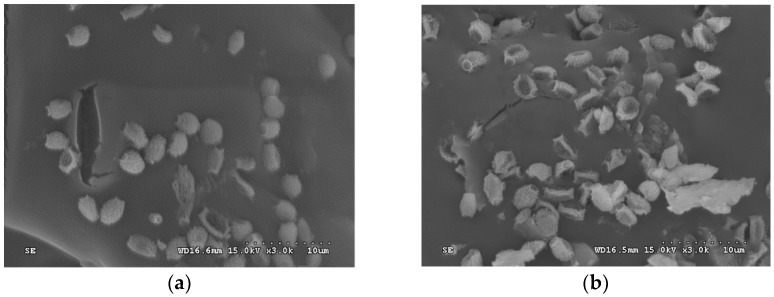
Scanning electron microscopy (SEM) images of inulin from (**a**) freeze-drying and (**b**) spray-drying.

**Table 1 molecules-24-01674-t001:** Drying response surface analysis test design and results (each test was done 3 times).

Run	Coded Variables			Actual Variables			Yield (%)
	A	B	C	X_1_	X_2_	X_3_	
1	0	1	−1	115	22	0.02	6.86 ± 0.03
2	0	0	0	115	20	0.03	7.49 ± 0.13
3	1	1	0	120	22	0.03	6.23 ± 0.06
4	−1	0	1	110	20	0.04	5.77 ± 0.02
5	−1	0	−1	110	20	0.02	7.35 ± 0.02
6	1	0	1	120	20	0.04	5.65 ± 0.04
7	0	1	1	115	22	0.04	5.31 ± 0.07
8	0	−1	1	115	18	0.04	5.97 ± 0.05
9	1	0	−1	120	20	0.02	6.58 ± 0.17
10	0	0	0	115	20	0.03	8.31 ± 0.11
11	0	−1	−1	115	18	0.02	6.48 ± 0.01
12	−1	1	0	110	22	0.03	6.56 ± 0.08
13	0	0	0	115	20	0.03	7.96 ± 0.10
14	0	0	0	115	20	0.03	8.03 ± 0.11
15	1	−1	0	120	18	0.03	7.03 ± 0.01
16	0	0	0	115	20	0.03	8.13 ± 0.15
17	−1	−1	0	110	18	0.03	6.13 ± 0.04

A, X_1_ represents temperature (°C); B, X_2_ are the number of creep speed(rpm) and C, X_3_ are the pressure (MPa).

**Table 2 molecules-24-01674-t002:** ANOVA analysis of experimental data.

Source	Sum of Squares	DF	Mean Square	F-Value	*p*-Value Prob > F	Significance
Model	13.35	9	1.48	15.56	0.0008	**
A	0.013	1	0.013	0.13	0.7248	N
B	0.053	1	0.053	0.55	0.4809	N
C	2.61	1	2.61	27.39	0.0012	**
AB	0.38	1	0.38	3.97	0.0866	N
AC	0.11	1	0.11	1.11	0.3274	N
BC	0.27	1	0.27	2.84	0.1360	N
A^2^	1.82	1	1.82	19.07	0.0033	**
B^2^	2.97	1	2.97	31.14	0.0008	**
C^2^	4.12	1	4.12	43.26	0.0003	**
Residual	0.67	5	0.095			
Lack of Fit	0.29	3	0.098	1.04	0.4645	N
Pure Error	0.37	4	0.094			
Cor Total	14.01	16				
R^2^	0.9524					
AdjustedR^2^	0.8912					

A represents temperature (°C); B represents the number of creep speed (rpm) and C represent the pressure (MPa). In the table above the ** means significant differences at *p* < 0.01, N: not significant at *p* > 0.05.

**Table 3 molecules-24-01674-t003:** Effect of drying method on inulin.

Method	Yield (%)	Water Content (%)	Particle Size (nm)
Freeze-drying	7.02 ± 0.56	4.34 ± 0.21	790.9 ± 80
Spray-drying	8.65 ± 0.69	3.49 ± 0.67	567.7 ± 37

**Table 4 molecules-24-01674-t004:** Coded levels of temperature, creep speed and pressure.

Variables	Code Level		
	−1	0	1
Heating temperature (A) °C	110	115	120
Creep speed (B) rpm	18	20	22
Pressure (C) MPa	0.02	0.03	0.04

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
