# Peer review of "Optimization of Spray-Drying Process of Jerusalem artichoke Extract for Inulin Production"

_molecules, 2019, doi:10.3390/molecules24091674_

Round 1

Reviewer 1 Report

The article entitled Optimization of spray-drying process of Jerusalem artichoke extract for inulin production is an interesting one. The authors present clearly the materials and method used for the extraction of inulin from Jerusalem artichoke but they do not explain the influence of the parameters on the extraction procedure.

The results and discussion section MUST BE improved regarding the scientific correlation between the parameters used for the extraction and the extraction yield.

Do the authors made some analysis for achieving the purity of the powder?

The conclusion section should be improved

Author Response

1.The article entitled Optimization of spray-drying process of Jerusalem artichoke extract for inulin production is an interesting one. The authors present clearly the materials and method used for the extraction of inulin from Jerusalem artichoke but they do not explain the influence of the parameters on the extraction procedure.

The results and discussion section MUST BE improved regarding the scientific correlation between the parameters used for the extraction and the extraction yield.

Do the authors made some analysis for achieving the purity of the powder?

Answer: The information about extraction yield was provided. The analysis of inulin purity was not carried out. We will make up the analysis in next season of Jerusalem artichoke for extraction tests.

2.The conclusion section should be improved

Answer: The conclusion was revised.

Reviewer 2 Report

The manuscript presents interesting results concerning spray drying of Jerusalem artichoke extract. These results might be useful for practical application. However, this manuscript requires improving due to some shortcomings. First of all better discussion of the results obtained is required.

Comments that should be considered to make the manuscript suitable for publication:

L 73. It is not clear whether the roots were prepared in some ways  before the extraction process for example by washing, cutting or smashing.

L 85-66. The ranges of variable process parameters are relatively small: (T, 110-120 °C), (V, 18-22 rpm) (P, 0.02-0.04 MPa). What was the reason for setting minimal and maximal values?

L 98. Probably  “freeze-dryer” should be replaced by “spray-dryer”. Provide the type of spry dryer as it is not clear what does it mean vacuum degree. Usually the pressure in the nozzle is specified. Please clarify this.

L 116-117. Consider replacing by “the difference in weight of two consecutive weightings was less than 2 mg”. What was the accuracy of the balance?

L 119. Define the abbreviations used in the equation. It should be the ratio of difference between the mass of fresh material and the mass of dried material to the mass of fresh material.

L 131. Equation (3) represents water content wet basis.

L 174. Specify, which table.

  178-179. “It is consistent with the particle size of inulin powders presented in Table 4.” Comment, in which way?

References require intensive editing for example: L 255-256. Correct the reference by adding an appropriate title of the journal.

Author Response

The manuscript presents interesting results concerning spray drying of Jerusalem artichoke extract. These results might be useful for practical application. However, this manuscript requires improving due to some shortcomings. First of all better discussion of the results obtained is required.

 Answer: The discussion was improved in the manuscript.

Comments that should be considered to make the manuscript suitable for publication:

1.L 73. It is not clear whether the roots were prepared in some ways before the extraction process for example by washing, cutting or smashing.

Answer: The information was added as "washed by tap water and sliced before use"

2.L 85-66. The ranges of variable process parameters are relatively small: (T, 110-120 °C), (V, 18-22 rpm) (P, 0.02-0.04 MPa). What was the reason for setting minimal and maximal values?

Answer: The ranges of variable process parameters were selected based on previous single factor tests. In the selected ranges, the inulin yield was relatively better.

3.L 98. Probably “freeze-dryer” should be replaced by “spray-dryer”. Provide the type of spry dryer as it is not clear what does it mean vacuum degree. Usually the pressure in the nozzle is specified. Please clarify this.

Answer: The type of spray dryer was provided.  ranges of variable process parameters were selected based on the operation limit of the spray-dryer and previous single factor tests. The vacuum degree was a mistake, we modified the expression.

4.L 116-117. Consider replacing by “the difference in weight of two consecutive weightings was less than 2 mg”. What was the accuracy of the balance?

Answer: The modification was done. The accuracy of the balance is 1 mg.

5.L 119. Define the abbreviations used in the equation. It should be the ratio of difference between the mass of fresh material and the mass of dried material to the mass of fresh material.

Answer: The modification was done.

6.L 131. Equation (3) represents water content wet basis.

Answer: Modification was done.

7.L 174. Specify, which table.

Answer: Modification was done.

8.178-179. “It is consistent with the particle size of inulin powders presented in Table 4.” Comment, in which way?

 Answer: The comment was modified to "It is consistent with the results that particle size of inulin powders from spray drying was more fine as presented in Table 4"

9.References require intensive editing for example: L 255-256. Correct the reference by adding an appropriate title of the journal.

Answer: modification was done.

Reviewer 3 Report

Comments for authors are in the attachment

Author Response

General remark

1. One of the basic property of powders is their solubility. It is a pity that the authors did not carry out the rehydration of powders. Perhaps the optimization of spray drying should have been done in this respect?

Answer: Great thanks to the comment about the solubility. To our knowledge, the solubility of inulin is limited at room temperature, it deed should be improved during production process. The present work considers more about the yield of the process. The respect of solubility will be considered in future work.

2.Freeze-drying was carried out at some of the parameters. What entitle authors to the assumption that these are the optimal conditions for freeze-dried of inulin?

Answer: The operation condition was selected according to our previous tests when freeze drying extract.

Detailed remarks

1.line 85

Please explain why a narrow temperature range has been used to optimize the process?

Answer: The ranges of variable process parameters were selected based on previous single factor tests.

2.line 90. Why yield was used as an indicator to optimize the process. Referring to equation (2), the more moist the powder, the higher the efficiency. Is this good?

Answer: Yield is important to evaluate if the process is efficient. We tested the water content of the inulin powder under the optimized condition, it is less than 5%.

3.line 98 It seems that there should be a “spray dryer”, not a “freeze dryer”. Please specify the model and manufacturer of the spray drier

Answer: The modification was done.

4. line 104 Please explain what the temperature is -50°C. Lyophilizer Alpha 2-4 does not have cooled shelves. If the temperature is due to the pressure used, please explain why this low pressure was used.

Answer: The freeze dryer used is Alpha 2-4 LD plus, the -50°C is the temperature of ice condenser.

5. Sublimation at very low pressure is very slow. And how intensively heat was supplied to the sublimation process? In which way it was stated, the process is completed?

Answer: The boiling point of the water decreases with the increase of the vacuum degree. The heating temperature of 110-120oC was used.

6. line 136. Please specify in how many repetitions each experiment was performed?

Answer: The information was provided in the text.

7.line 174. In which vessels were the inulin contained in the freeze dryer? Maybe, you had to use dishes with non-stick coatings, and then the yield would be higher?

Answer: The plastic petri dish used for freez-drying. We will improve our methods in future study according to the comments.

8. line 176. Humidity at the level of 3 to 5% is so low that it does not differentiate the shelf life

Answer: The sentence about shelf life was deleted.

9. Table 4. What is “Particle size (nm) is this the average value or the dominant particle size? It was necessary to give the particle size distribution. This would allow to compare the particle size of both powders.

Answer: Thanks for the precious suggestion of the reviewer. The particle size is the average value. We will make up the particle size distribution analysis in next season of Jerusalem artichoke.

10. Figure 3. Please enter the zoom at which the images were taken

Answer: The information was added in the figure.

Reviewer 4 Report

I my opinion this article needs major revision. Protocol sound good and introduction also, however, almost no disscusion is made.

Authors should disscuss all results that they obtained.

best wishes,

R.

Author Response

I my opinion this article needs major revision. Protocol sound good and introduction also, however, almost no disscusion is made. 

Authors should disscuss all results that they obtained. 

Answer: More discussions were added in the text.

Reviewer 5 Report

The manuscript entitled: ‘Optimization of spray-drying process of Jerusalem artichoke extract for inulin production’ is on the optimization spray drying of inulin from Jerusalem artichoke using RSM. The inulin yields have been only considered. However, the presence of inulin in the extracts should have been confirmed by the use of a spectrophotometer or HPLC. In my opinion the data are not enough for publication in Molecules.  Please find specific comments below.

·       Lines 51-54: Reference should be added.

·       Explanation how the levels of pray drying conditions were selected for optimization should be given.

·       The statistical analysis section must be added in the materials and methods.

·       In Tables 2 and 3 what does each letter mean should be explained below the table.

·       Comparison between freeze and spray drying? Are the values significantly different please explain? If yes at what level the comparison was mad and how?

·       Was inulin determined any spectrophotometric methods? At least it should have been confirmed the presence of inulin in the extract by HPLC o spectrophotometer.

·       SEM figures should be of higher magnification especially for spray drying.

Author Response

The manuscript entitled: ‘Optimization of spray-drying process of Jerusalem artichoke extract for inulin production’ is on the optimization spray drying of inulin from Jerusalem artichoke using RSM. The inulin yields have been only considered. However, the presence of inulin in the extracts should have been confirmed by the use of a spectrophotometer or HPLC. In my opinion the data are not enough for publication in Molecules.  Please find specific comments below.

1. Lines 51-54: Reference should be added.

Answer: References were added.

2.Explanation how the levels of pray drying conditions were selected for optimization should be given.

Answer: the selection of the levels was based on previous single factor experiments.

3.The statistical analysis section must be added in the materials and methods.

Answer: the statistical analysis section was added.

4. In Tables 2 and 3 what does each letter mean should be explained below the table.

Answer: Modification was done.

5.Comparison between freeze and spray drying? Are the values significantly different please explain? If yes at what level the comparison was mad and how?

Answer: Explanation for difference analysis was added in " 2.5. Statistical Analysis"

6.Was inulin determined any spectrophotometric methods? At least it should have been confirmed the presence of inulin in the extract by HPLC o spectrophotometer.

Answer: Inulin test was done in our previous work. "Development of a Combined Trifluoroacetic Acid Hydrolysis and HPLC-ELSD Method to Identify and Quantify Inulin Recovered from Jerusalem artichoke Assisted by Ultrasound Extraction, Applied Sciences, 2018, 8, 710."

7. SEM figures should be of higher magnification especially for spray drying.

Answer: The new SEM figures were provided as required.

Round 2

Reviewer 1 Report

In my opinion the authors made the requiered requests and I believe that the article should be accepted as it is.

Reviewer 2 Report

In my opinion, the manuscript can be published in its current form.

Reviewer 4 Report

I accept this article.

Reviewer 5 Report

The authors kindly replied to the comments made by this reviewer. The level of significance where the statistical analysis was made should be added in the statistical analysis section. I assume it was p<0.05?